REGISTERED REPORT PROTOCOL

# Effects of whole-body electromyostimulation on function, muscle mass, strength, social participation, and falls-efficacy in older people: A randomized trial protocol

Túlio Medina Dutra de Oliveira[1], Diogo Carvalho Felício[1‡], José Elias Filho[1,2], João Luiz Quagliotti Durigan[3‡], Diogo Simões Fonseca[1], Anderson José [1], Cristino Carneiro Oliveira[1‡], Carla Malaguti[1‡]*

1 College of Physiotherapy, Federal University of Juiz de Fora, Juiz de Fora, Minas Gerais, Brazil, 2 College of Physical Education and Sports, Federal University of Juiz de Fora, Juiz de Fora, Minas Gerais, Brazil, 3 College of Physiotherapy, University of Brasília, Brasília, Distrito Federal, Brazil

☯ These authors contributed equally to this work.
‡ These authors also contributed equally to this work.
* carlamalaguti@gmail.com

## Abstract

### Background

Resistance training has a positive impact on functional capacity and muscle mass in the elderly. However, due to physical limitations or a simple aversion against regular exercise, a majority of the elderly do not reach the recommended exercise doses. This led us to evaluate the effect of whole-body electromyostimulation (WB-EMS), a novel, time-efficient, and smooth training technology on physical function, fat-free mass, strength, falls-efficacy, and social participation of the elderly.

### Methods

The present study is a randomized, parallel group clinical trial approved by the Ethics Committee of our Institution. Sixty-six volunteers (age ≥ 60 years) will be recruited from the geriatric outpatient department in a tertiary hospital and primary care units and randomized into two groups: WB-EMS group or active control group (aCG). The WB-EMS or aCG protocol will consist of 16 sessions for 8 consecutive weeks, twice per week. The primary outcomes will be maximal isometric knee extension (IKE), functional lower extremity strength, fat-free mass, gait speed, and risk of falls measured before and after intervention. The secondary outcomes will be social participation and falls-efficacy assessed before and after the intervention and at three and six months of follow-up. Participant's satisfaction with and awareness of electrical stimulation therapy will also be assessed immediately after the 8-week intervention.

### Discussion

Patients receiving WB-EMS exercises are believed to have better outcomes than those receiving conventional, more time-consuming resistance exercises. Hence, innovative,

---

**Data Availability Statement:** All relevant data from this study will be made available upon study completion. Data are available from the Research Ethics Committee of the Federal University of Juiz de Fora (Rua Catulo Breviglieri, s / n˚ - Santa Catarina, Juiz de Fora, Brazil - CEP: 36036-110 telephone contact +55 (32) 4009-5167) for researchers who meet the requirements for access to confidential data.

**Funding:** This study was partially funded by the FAPEMIG - number APQ-0354-17 –Universal Demand 001/2017 and Coordenação de Aperfeiçoamento de Pessoal de Nível Superior - Brasil (CAPES) – Finance Code 001.

**Competing interests:** The authors have declared that no competing interests exist.

time-efficient, joint-friendly, and highly individualized exercise technologies (such as WB-EMS) may be a good choice for the elderly with time constraints, physical limitations, or little enthusiasm, who are exercising less than the recommended amounts for impact on muscle mass, strength, and function.

## Introduction

An aging population is a prominent worldwide phenomenon. In the past few decades, developing countries have shown a progressive decline in mortality rates and, more recently, in their fertility rates as well, which has resulted in an increase in the older population [1]. The proportion of the world's population over 60 years of age doubled from 12% (900 million people) to 22% (2 billion) between 2015 and 2050 [2]. Aging is accompanied by an increase in the prevalence of chronic degenerative diseases and comorbidities, reflecting the decrease in functional capacity, quality of life, and autonomy [3, 4]. Other changes due to aging further increase the progression of sarcopenia [5].

The European Working Group on Sarcopenia in Older People defined sarcopenia as a progressive and generalized musculoskeletal disorder, which intensifies after the age of 50, with a 1.5% - 5% annual loss of muscle strength, associated with a higher probability of risk of falls, fractures, physical disability, and mortality [6–9]. It is defined as primary sarcopenia, or age-related, when no other specific cause is evident. It is termed secondary sarcopenia when other causal factors are associated with aging [6]. Although secondary factors, such as lifestyle and physical inactivity, can potentiate functional disability and loss of strength, interventions such as resistance training seem to delay or reverse this process [10,11].

However, due to the physiological principle of reversibility, gains in muscle strength and endurance due to extensive resistance training can be lost with the discontinuation of exercises [12]. Older people tend to participate more assiduously at the beginning of exercise programs and become less enthusiastic over time [13]. Studies suggest that 50% of people who begin a resistance training program drop out within six months [14, 15]. Reasons for the lack of adherence to the training program in the older population include pain, difficulty in performing the exercises, poor motivation, lack of professional supervision, and fear of falling [16]. Hence, new training strategies should be optimized to improve adherence to therapeutic programs in the elderly [16]. Whole-body electromyostimulation (WB-EMS) has recently been used as a resistance training option [17].

WB-EMS is based on the same mechanisms of action as classical neuromuscular electrical stimulation (NMES), which is only applied locally. However, WB-EMS can be used with several electrodes at the same time and positioned in different muscle groups to cover an area of up to 2,800 cm$^2$, globally combining electrical myostimulation with functional movements [18]. One of the advantages of WB-EMS is that it acts directly toward the synthesis of skeletal muscle proteins and is faster than conventional techniques. Some studies have shown that 18 min of training, twice a week for 12 months, increased appendicular muscle mass, strength, and decreased abdominal fat mass [17, 19]. In addition, WB-EMS was feasible, had high adherence and low dropout rates among the elderly [17, 19]. Despite demonstrating promising results, few studies have investigated the effectiveness of WB-EMS in the elderly or patient outcomes related to functional capacity and follow-up [19, 27, 28]. Studies involving WB-EMS in the elderly are scarce and only assessed structural as body composition, strength, or laboratory outcomes as lipid profiles [19, 29, 44]. Little or no studies using WB-EMS involving older people assessed outcomes of functional capacity and related to the falls-efficacy and social

participation in these patients. In addition, no study of WB-EMS with the older people had follow-up and longitudinal assessment of the maintenance of its effect.

The primary aim of this study is to assess the immediate clinical effects of an 8-week WB-EMS exercise program on maximal isometric knee extension (IKE), functional lower extremity strength, fat-free mass, gait speed, and risk of falls. The secondary aim will assess the immediate, medium (three months), and long-term (six months) effectiveness of WB-EMS on social participation and falls-efficacy of sedentary older people. Furthermore, participant's satisfaction with and awareness of electrical stimulation therapy will also be assessed immediately after the 8-week intervention. The hypothesis of this trial is as follows: the WB-EMS-associated voluntary exercise protocol is more effective for improving strength, increasing lean mass, and modulating the functional aspects related to the effectiveness of falls and social participation compared to the control group training with resistance exercises.

## Methods

### Study design

This is a protocol for a clinical, randomized controlled, parallel, single-blinded trial (outcome assessors). This trial was designed according to the Standard Protocol Items: Recommendations for Intervention Trials (SPIRIT) statement (**S1 Appendix**) [20] and will be reported according to the Consolidated Standards of Reporting Trials (CONSORT) statement [21]. Two different exercise groups will be formed through stratified random sampling: an experimental group (EG), which will undergo WB-EMS training, and an active control group (aCG), which will undergo resistance exercise training. Assessment will occur at baseline and after eight weeks, three months, and six months after intervention. Assessments will be conducted by an independent assessor who will be blinded to the group allocation. At the end of the 16 treatment sessions, the primary and secondary outcomes of the study participants will be reassessed by the same evaluator who performed the baseline assessments. Falls-efficacy and social participation will be monitored monthly between the third and sixth month. The progress through the phases of enrollment, intervention allocation, and follow-up are shown in **Fig 1**. The protocol of this study has been registered at the Brazilian Clinical Trials Registry (RBR-422x64) and approved by the Research Ethics Committee of Federal University of Juiz de Fora (3.889.143).

Eligible participants will be informed about the objectives, risks, and benefits of the study assessors and will be required to complete the informed consent form according to the Brazilian National Health Council Resolution 580/2018. To ensure the privacy and confidentiality of the data collected, all research personnel will take appropriate measures. The confidentiality of the information will be protected by the principal investigator who will omit the information on the identification of the participants by means of codes and restriction of access to electronic files by the exclusive use of a password. The data collected and analyzed by this project will be disseminated at congresses and through international peer-reviewed journals and will not be reported in any of the forms of dissemination in this study.

### Sample size calculation

Sample size was calculated using the t-test for independent groups using G*Power 3.0.10 software (University of Kiel, Kiel, Germany). All primary outcome variables with $\alpha = 5\%$ and 80% statistical power were considered to calculate the sample size. Sample size for maximal voluntary contraction to detect a difference between groups with an effect size of 0.96 assumed [22], which generated a sample size of 36 (18 participants per group). For functional lower extremity strength, an effect size of 0.38 was assumed [23], generating a sample size of 57 (29 participants

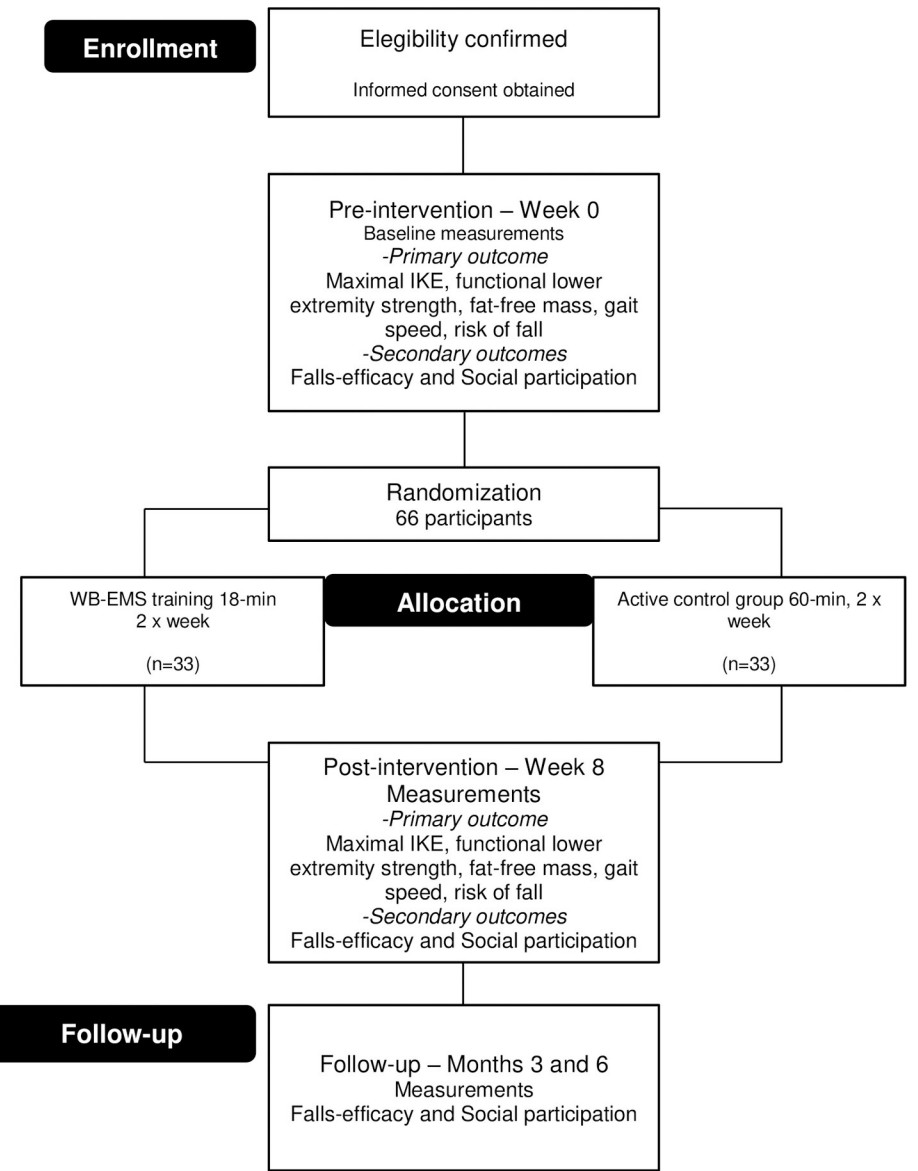

**Fig 1. CONSORT flowchart of the planned protocol pathway.** IKE, isometric knee extension; WB-EMS, whole-body electromyostimulation.

per group). For fat-free mass, a difference of 1.12 kg between groups and a standard deviation of 0.79 kg was assumed [24], generating a sample size of 26 (13 participants per group). For gait speed, a clinically meaningful change was considered to be small when an improvement of 0.05 m/s was present, and considerable when an improvement of 0.10 m/s was present [25], generating a sample size of 40 (20 participants per group). For risk of fall, a difference of 21.3% between groups and a standard deviation of 3.5 seconds were assumed [26], generating a sample size of 14 (7 participants per group). Thus, to ensure suitable power and assuming sample losses, 68 participants (34 participants per group) will be considered (assuming attrition of five subjects per group– 15%) based on functional lower extremity strength outcome, that provides the largest sample size.

## Participants

The participants will be sedentary, greater than 60 years, recruited from basic health units and the geriatric clinic of the university hospital and health care center of the Federal University of Juiz de Fora (HU-CAS/ UFJF). The inclusion and exclusion criteria are outlined in **Table 1**.

## Recruitment method and screening procedures

Patients in the geriatric outpatient clinic on the waiting list and who meet the inclusion criteria of the study will be recruited through phone contact, email, or in person. The disclosure will also be made through recruitment flyers at the hospital, university, and through social networks, guiding those participants who consider themselves eligible to contact a researcher who is part of the study. By reading and signing the informed consent form (**S2 Appendix**), participants will provide their consent to participate in the trial. After this, a physical therapist blinded to the group allocation will collect baseline data to confirm eligibility. This assessment will be conducted at the Laboratory of Movement Analysis of Physiotherapy Faculty of the Federal University of Juiz de Fora.

## Randomization and allocation

Once patients have accepted an invitation to participate in the trial, they will provide their written consent before being assessed for eligibility. Participants eligible for the study will be divided into two strata based on sex. Stratified randomization with a block size of two will be used to assign patients to groups 1 and 2. Randomization will be performed by a 1:1 allocation ratio to the EG or aCG using Random Allocation Software (RAS). The allocation sequence will be kept in sequentially numbered identical opaque sealed and stapled envelopes and will be kept hidden until the end of the evaluation. The envelope will be sealed with tamper-resistant labels and lined with aluminum foil, making it impervious to intense light. The researcher will be instructed not to disclose the groups to the therapist or other researchers involved in the study until completion.

## Blinding

Due to the nature of this study, it is not possible to fully blind the patient or the clinician providing the intervention to the treatment received. All outcome measures collected at baseline, 8-week, and follow-up assessments will be evaluated by an assessor who will not know the identities of the participants in the allocated treatment group.

**Table 1. Inclusion and exclusion criteria.**

| Inclusion criteria | Exclusion criteria |
| --- | --- |
| 1. Age 60–80 | 1. Cognitive alterations detectable by the Mini Mental State Examination[1] |
| | 2. Participation in structured physical activity in last year |
| | 3. Uncontrolled cardiac illness and/or metabolic disease |
| | 4. Known history of cerebrovascular disease or sequelae neurological |
| | 5. Active neoplasia during the previous five years |
| | 6. Surgical fractures or osteosynthesis in the last six months |
| | 7. Severe visual and hearing difficulties |
| | 8. Other pathological or orthopedic conditions that limits physical training or assessment outcomes |

[1] Folstein MF, Folstein SE, McHugh PR. Mini-Mental State: a practical method for grading the cognitive state for the clinician. J Psychiatr Res 1975, 12: 189–198.

**Table 2. Whole-body electromyostimulation protocol.**

| WB-EMS protocol |
|---|
| Stimulation frequency: 85Hz |
| Impulse duration: 5s |
| Impulse break: 10s |
| Pulse duration: 350 μs |
| Impulse type: bipolar |
| Duration: 18 min |

## WB-EMS intervention

The WB-EMS will be used to simultaneously activate 8–12 muscle groups (upper legs, upper arms, bottom, abdomen, chest, lower back, upper back, and latissimus dorsi) with different levels of intensity. The bipolar electric current by WB-EMS devices from Miha bodytec® (Gersthofen, Germany) will be initially applied with the following parameters: frequency of 85 Hz and pulse duration of 350 μs, intermittent with 5 s of EMS stimulation to perform the movement and 10 s of rest **(Table 2)**. The current intensity will be individually selected and modified during the same EMS session. The WB-EMS protocol will be applied based on a low-intensity/low-amplitude movement protocol according to settings described in previous studies [27–29]. After performing a 5-minute warm-up, participants will undergo 18 minutes of supervised WB-EMS training twice a week on alternating days, until 8 weeks passed. The 8-week intervention period comprised of 16 training sessions. Groups of two participants will be supervised by an instructor; the session will also be acoustically and visually guided by videos that demonstrate the movements of the protocol. The WB-EMS protocol will closely follow the setting of commercial WB-EMS sessions with their low-loading/low-amplitude movement strategies. **Table 3** presents the "core exercises" composed of five basic movements that will be combined, generating ten dynamic exercises that will be performed in an orthostatic position without the addition of weights [29]. The WB-EMS training will be structured in 1–2 sets of 6–8 repetitions. Low speed amplitude and intensity movements will be prescribed (squat: leg-flexion < 35˚) to avoid the effects of the exercise itself. Moreover, no progressive increments of the exercises will be applied during the study. After the adaptation period of four WB-EMS sessions, the current intensity will be individually adjusted according to the tolerance level of participants during the same session. Due to stimulated sites' impedance differences, the participants maintain a rate of perceived exertion (RPE) of "hard" to "very hard" (Borg CR-10 Scale "6" of "10") [30] during the session. The current intensity could be a key element for positive effects compared to conventional exercise programs, so more emphasis should be given to this parameter. The corresponding current intensity will be saved for each region on chip cards to generate a fast, reliable, and valid setting during the subsequent WB-EMS sessions.

**Table 3. Exercises performed under WB-EMS application [29].**

| Exercises protocol: |
|---|
| 1. Deadlift (6 s down) with arm extension/deadlift (6 s up) with arm flexion |
| 2. Squat (6 s down) with trunk flexion (crunches) |
| 3. Squat (6 s down) with lateral pulleys/squat (6 s up) with military press |
| 4. Squat (6 s down), crunch with butterfly/squat (6 s up) and reverse fly |
| 5. Squat (6 s down) and vertical chest press/squat (6 s up) and vertical rowing |

Hz: Hertz; s: second; μs: microsecond; min: minutes.

The physiotherapist will monitor the interventions, answer questions, and supervise the exercise performance during the program. In each session, participants will be examined for adverse signs and symptoms such as increased pain, extreme discomfort, and intolerance to exertion. The activity will be stopped if the participants reach level 8 or more of dyspnea or fatigue on the Borg scale. If any soreness persists for more than a few hours after the intervention, the intensity will be decreased in the next session for that participant.

## Active control group

The aCG training will be carried out at the fitness gym of the Physical Education Faculty of the Federal University of Juiz de Fora. Each training session will consist of a 10-minute warm-up with walking and movement of different body parts: arms, wrists, fingers, shoulders, legs, and ankles. The resistance training will consist of an 8-week guided training on fitness devices (pull down, leg press, bench press, back press, etc.) involving all large muscle groups. Participants will take part in the resistance training program for two sessions per week, 60 min each. Individual adaptations of the training protocol will be made regularly as a function of the actual performance. The intensity will be based on the number of possible repetitions (weeks 1–2: 15 repetitions, weeks 2–6: 9 repetitions), with an intensity of 50%– 60% of 1-repetition maximum (1-RM) in the first two weeks, and later with 70%–80% 1-RM. A training volume of three sets per exercise and rest intervals of 4 s between repetitions and 60 s between sets will be defined. Participants will be stimulated to maintain muscle tension for 6 s [31].

All participants from both groups will receive a weekly call with questions about their general health as well as reinforcement about the day and time of their respective physical training.

## Primary outcomes

The primary measured outcomes are: maximal IKE strength, functional lower extremity strength, fat-free mass, gait speed, and risk of falling. They will be measured at baseline and immediately after the 8-week intervention period.

Maximal IKE strength will be measured by the MicroFET® (Hoggan Health Industries, West Jordan, UT, USA) handheld dynamometer [32]. All force measurements will be acquired using isometric tests, where patients will be seated with their legs vertical and the dynamometer applied perpendicular to the leg proximal to the malleoli. This measurement will be performed five times, with the highest and lowest values being discarded. The average of the three remaining values will be calculated, and registered in newtons.

Functional lower extremity strength will be measured using a 30-second sit-to-stand test. It will be performed in a standard chair (height; 44 to 45 cm) with no arm support. Participants will be instructed to stand up from and sit down on the chair with no support from the hands, repeating the procedure as many times as possible for 30 s [33]. The test will be first demonstrated by an evaluator and then performed by the participant. The number of stands will be recorded manually. The 30-second sit-to-stand test has acceptable reliability when testing older people [34].

Fat-free mass will be assessed by bioelectrical impedance analysis (Biodynamics 310, Biodynamics Corporation, Seattle, WA, USA). All body composition measurements will be taken at the same time of day. During the measurements, participants will be laid in a supine position with their limbs slightly apart from their bodies. Two electrodes will be positioned on the dorsal surface of the right hand, and two additional electrodes will be positioned on the dorsal surface of the right foot. The fat-free mass (FFM) will be calculated using an equation: (FFM

index = body weight (in kg) of FFM/height (in m) squared) to adjust for body surface area [35].

Maximal and preferred gait speed will be assessed using the distance/time (m/s) ratio, measured across ten meters. The gait speed will be recorded only in the central six meters of the track, identified laterally by tape marks, to avoid acceleration and deceleration bias. Participants will be instructed to stand with both feet behind the start line and to start walking after a specific verbal command. The tests will be repeated three times to yield an average maximal and preferred gait speed [36].

The risk of falling will be measured using the timed up and go test (TUG). Participants will be asked to get out of the chair, walk three meters, turn around, walk back to the chair, and sit down, assisted by a go signal. In each measuring session, the TUG test will be repeated five times (five trials/session). To avoid falls during the tests, patients will be instructed to use a comfortable walking speed. Participants will have one practice trial, and the second trial will be timed. The TUG test has demonstrated good accuracy in the prediction of falls among the elderly [37].

The same research outcome assessors, blinded to the status of the participants, will perform the tests at baseline and post-intervention and will be responsible for conducting data collection. Follow-up data will be collected by the research assistant by phone or mail. The assessor will assess outcomes in participants who dropped out of the study.

## Secondary outcomes

Secondary outcomes to be measured are social participation, falls-efficacy, and participant's satisfaction.

Social participation will be measured using the Assessment of Life Habits (LIFE-H) questionnaire [38]. It comprises 69 life habits across 12 categories. These categories (number of items) are nutrition, fitness, personal care, communication, housing, mobility, responsibilities, interpersonal relationships, community life, education, employment, and recreation. The first six categories refer to daily activities, while the others are associated with social roles. In the present study, because of their irrelevance for the majority of the elderly, the categories "employment" and "education" were removed from analysis, leaving 10 categories and 59 items. This questionnaire was culturally adapted and translated to Brazilian Portuguese [39].

Falls-efficacy will be measured using the Falls-Efficacy International Scale—Brazil (FES-I-BRAZIL), adapted and validated for the Brazilian population [40]. Scores can range from 16 (with no concern for falling) to 64 (with extreme concern). The cut-off point for fear of falling will be a score of 23, as cited in the literature [41].

Participant's satisfaction with and awareness of electrical stimulation therapy will be examined by the patients' satisfaction with and awareness of electrical stimulation therapy instruments [42]. This questionnaire includes two sections. Section 1, consisting of ten items, addressing demographic details such as age, gender, education level, application of body areas, number of treatment sessions, electrical stimulation therapy (EST) modalities, perceived healing effect, devices, and probes of EST. Data will be collected via closed-ended categorical and yes/no questions. Section 2, consisting of three items, investigating the participants' having information on EST, knowledge of EST, and satisfaction. The questionnaire will be administered face-to-face to the volunteers. The scored questions will be analyzed as percentiles.

## Monitoring of data quality

To ensure data quality, the research assistant who collects the data sheets, also will provide feedback to the principal researcher if there is evidence that the protocol is not being followed.

Data will be entered and double-checked by two people, and inconsistencies resolved by contacting the participant where appropriate or via consensus.

## Data analysis

The statistical analysis of primary and secondary outcome measurements will include all randomized patients analyzed within their original groups by intention-to-treat. Data normality will be analyzed using the Kolmogorov-Smirnov test. Parametric data will be represented as means (SD) and non-parametric data as medians (IQR, $25^{th}$– $75^{th}$ percentiles). A two-way repeated measurement by ANOVA will be conducted with "Time" (two levels: pre-intervention, post-intervention) and "Groups" (WB-EMS and Control) for primary outcomes and "Time" (four levels: pre-intervention, post-intervention, three and six-month follow-up) and "Groups" (WB-EMS and Control) for secondary outcomes. Their respective 95% CIs will be calculated using mixed linear models [43]. The percentage of missing data, effect size, and other non-normal distributions will be considered as criteria for covariance structures in the mixed linear model [44, 45]. Power calculation will be performed *a posteriori*, and effect sizes will be determined using partial eta squared ($\eta_{\rho}^{2}$). Cohen (1988) provided benchmarks to define small ($\eta_{\rho}^{2} = 0.01$), medium ($\eta_{\rho}^{2} = 0.06$), and large ($\eta_{\rho}^{2} = 0.14$) effects. A value of $p < 0.05$ will be set as significant. SPSS version 13 (Chicago, IL, USA) will be used as the statistical software for analysis. All relevant data will be added within the paper and its Supporting Information files.

## Discussion

This manuscript describes the rationale and processes of a study investigating the effectiveness of implementing a WB-EMS exercise program to evaluate health indicators of the elderly. Although conventional resistance exercise is the most recommended intervention for the management and prevention of sarcopenia, time constraints, physical limitations, and low motivation to engage in an unsupervised exercise program can often be a problem [13, 16]. These issues are particularly pronounced for older populations who are more likely to have impaired physical performance, loss of muscle mass and strength due to the senescence, and poor adherence to exercise programs [15, 46].

WB-EMS exercises present an opportunity to increase adherence to an exercise program in the elderly, since it is a less time-consuming therapy than conventional resistance exercises [29]. Recently uncovered evidence has revealed that WB-EMS may be a beneficial alternative to conventional physical exercise in different populations, especially in the elderly [17]. Recent studies have also shown that this new technology is feasible and effective for older people and is a favorable option for improving body composition and physical strength in postmenopausal and overweight women [19, 29]. However, these studies did not evaluate mobility functions or whether the effects were maintained longitudinally [19, 29]. In addition, most of these studies were concentrated in Germany, which makes it difficult to generalize the results, considering that the health conditions of this cohort and the cultural specificity could affect the results. [19, 29, 47].

## Supporting information

**S1 Appendix. SPIRIT checklist.**
(DOC)

**S2 Appendix. Consent form.**
(DOCX)

**S1 File.**
(DOCX)

**S2 File.**
(DOCX)

## Author Contributions

**Conceptualization:** Túlio Medina Dutra de Oliveira, Diogo Carvalho Felício, José Elias Filho, João Luiz Quagliotti Durigan, Diogo Simões Fonseca, Anderson José, Cristino Carneiro Oliveira, Carla Malaguti.

**Investigation:** João Luiz Quagliotti Durigan, Diogo Simões Fonseca, Anderson José.

**Methodology:** Túlio Medina Dutra de Oliveira, Diogo Carvalho Felício, José Elias Filho, Diogo Simões Fonseca, Anderson José, Cristino Carneiro Oliveira, Carla Malaguti.

**Project administration:** Anderson José, Carla Malaguti.

**Resources:** Diogo Carvalho Felício.

**Supervision:** José Elias Filho.

**Writing – original draft:** Túlio Medina Dutra de Oliveira, Diogo Carvalho Felício, José Elias Filho, João Luiz Quagliotti Durigan, Diogo Simões Fonseca, Anderson José, Cristino Carneiro Oliveira, Carla Malaguti.

**Writing – review & editing:** Diogo Simões Fonseca, Cristino Carneiro Oliveira.

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
