## [Decision Letter · Decision Letter 0]

13 Nov 2020

PONE-D-20-23831

Effects of whole-body electromyostimulation on function, muscle mass, strength, social participation, and falls-efficacy in older people: A randomized trial protocol

PLOS ONE

Dear Dr. MALAGUTI,

Thank you for submitting your manuscript to PLOS ONE. After careful consideration, we feel that it has merit but does not fully meet PLOS ONE’s publication criteria as it currently stands. Therefore, we invite you to submit a revised version of the manuscript that addresses the points raised during the review process.

The reviewers had some minor concerns about the methods that need to be addressed prior acceptance, particularly the choice of control group needs to be better supported.

We look forward to receiving your revised manuscript.

Kind regards,

Cameron J. Mitchell, PhD

Academic Editor

PLOS ONE

Journal Requirements:

3. Thank you for submitting the above manuscript to PLOS ONE. During our internal evaluation of the manuscript, we found significant text overlap between your submission and the following previously published work:

https://link.springer.com/article/10.1007%2Fs11357-013-9575-2

https://www.hindawi.com/journals/ecam/2016/9236809/

https://www.physiotherapyjournal.com/article/S0031-9406(16)30024-4/fulltext

https://journals.plos.org/plosone/article?id=10.1371/journal.pone.0221937

Please revise the manuscript to rephrase the duplicated text, cite your sources, and provide details as to how the current manuscript advances on previous work. Please note that further consideration is dependent on the submission of a manuscript that addresses these concerns about the overlap in text with published work.

4. There is a discrepancy in the number of participants between the manuscript/ protocol and registry (http://www.ensaiosclinicos.gov.br/rg/RBR-422x64/). Please clarify why this is so.

"This study is partially funded by FAPEMIG - number APQ-0354-17 –

 Universal Demand 001/2017 and Coordenação de Aperfeiçoamento de Pessoal

de Nível Superior - Brasil (CAPES) – Finance Code 001."

"No"

7. Your ethics statement should only appear in the Methods section of your manuscript. If your ethics statement is written in any section besides the Methods, please delete it from any other section.

Reviewers' comments:

Reviewer's Responses to Questions

**Comments to the Author**

1. Does the manuscript provide a valid rationale for the proposed study, with clearly identified and justified research questions?

Reviewer #1: Yes

Reviewer #2: Partly

Reviewer #3: Yes

2. Is the protocol technically sound and planned in a manner that will lead to a meaningful outcome and allow testing the stated hypotheses?

Reviewer #1: Yes

Reviewer #2: Yes

Reviewer #3: Yes

3. Is the methodology feasible and described in sufficient detail to allow the work to be replicable?

Reviewer #1: Yes

Reviewer #2: Yes

Reviewer #3: No

4. Have the authors described where all data underlying the findings will be made available when the study is complete?

Reviewer #1: Yes

Reviewer #2: Yes

Reviewer #3: No

5. Is the manuscript presented in an intelligible fashion and written in standard English?

Reviewer #1: Yes

Reviewer #2: Yes

Reviewer #3: Yes

6. Review Comments to the Author

You may also provide optional suggestions and comments to authors that they might find helpful in planning their study.

Reviewer #1: The investigators propose to conduct a randomized controlled clinical trial with a target accrual of 66. Individuals will be assessed at four time points.

Minor revisions:

1- Abstract, Background: Typographical error: the majority of older people…

2- Line 155: Sample size calculation: Indicate the statistical testing methods which achieves the state power estimates.

3- Line 367: Should time have 4 levels since assessments will be collected at 4 time points?

4- Line 369: State the criteria that will be used to select the underlying covariance structure in the mixed linear model.

Reviewer #2: This manuscript describes the protocol of an 8-week RCT on the effect of whole-body electrical stimulation versus a resistance exercise program in older adults. The study includes a comprehensive set of measures of muscle mass, strength, and function, as well as secondary outcomes related to social participation and falls efficacy. The study protocol seems well thought-through, but I do have the following comments:

Major comments:

1) My main comment is about the aim of the study, based on the hypotheses (line 117): is it really expected that the ‘active control group’ will show greater reductions in strength and FFM than the WB-EMS group? I understand that the longer-term compliance to a RE program is not large, but I find it hard to believe that WB-EMS will have a greater effect on muscle function and strength than supervised resistance exercise. Would it not be better to compare the WB-EMS group with a true control group, doing habitual physical activity only?

2) What gap in the literature will this study fill? What will this study demonstrate that the studies in line 111 haven’t?

3) The sample size calculation is currently unclear: is the study powered to pick up a change in variables in the WB-EMS group, or a difference between both intervention groups?

Minor comments:

1) With n=29 participants per group required for functional lower extremity strength, and a dropout of 5 participants per group, should the sample size not be n=34 per group?

2) Line 79: should this be ‘annual loss’?

3) How will the four free options be decided? (line 218)

4) Line 307: time of day?

5) Line 367: pre-intervention instead of pre-WB-EMS

6) Line 398: why poor adherence to long-term programs relevant for such a short study?

Reviewer #3: Thank you for the opportunity to review this manuscript. In general, it is well written and well designed.

I have just a few comments regarding the methods.

1- I believe that the word “sex” is more appropriated than “gender”.

2- The randomization method should be better described. The authors describe the allocation concealing method with opaque envelopes, which is appropriated. However, how will the random allocation be performed? Will the authors use some software or website for this? Or will the authors draw envelopes? Please clarify.

3- “Pulse width” should be replaced by “pulse duration” as it is measured in time unit.

4- The current amplitude/intensity adjust should be better described. Please give more details. Will the amplitude be increased during the same treatment session?

5- Why the ON time will be longer than the OFF time 6/4? In traditional NMES programs for increase of muscle strength the OFF time is longer than the ON time to avoid fatigue. Please clarify.

6- Why did authors decided to perform 2 sessions per week rather than 3? Are 2 sessions per week an appropriated dose for exercise/electrical stimulation?

7- In the Primary and Secondary Outcomes section the timepoint (s) for each outcome should be described.

8- I suggest some assessment regarding patient’s satisfaction. In introduction authors describe the lack of adherence in exercise programs and suggest the whole-body electromyostimulation as a new training strategy to improve older people’s adherence to the therapeutic programs. Therefore, I believe it is very important to assess the patients’ impressions on this new method compared to the traditional (control group).

7. PLOS authors have the option to publish the peer review history of their article (what does this mean?). If published, this will include your full peer review and any attached files.

Reviewer #1: No

Reviewer #2: No

Reviewer #3: No

---

## [Author Response · Author response to Decision Letter 0]

17 Dec 2020

We would like to thank the reviewers for their thoughtful comments and efforts towards improving our manuscript. Our response to the Reviewer’s comment is described point-to-point below. Appropriated changes suggested by the Reviewers have been introduced to the manuscript (highlighted within the document).

Reviewer #1: The investigators propose to conduct a randomized controlled clinical trial with a target accrual of 66. Individuals will be assessed at four time points.

Minor revisions:

1- Abstract, Background: Typographical error: the majority of older people…

Answer: We apologize for our typographical error. We have already corrected it in the manuscript.

2- Line 155: Sample size calculation: Indicate the statistical testing methods which achieves the state power estimates.

Answer: We used the t-test for independent groups through the G*Power 3.1.9.6 software. We added this information in the Sample Size Calculation section.

3- Line 367: Should time have 4 levels since assessments will be collected at 4 time points?

Answer: We have four assessment levels for secondary outcomes, so we added the following information in the sentence: 

 “A two-way repeated-measures ANOVA will be conducted with “Time” (2 levels: pre-intervention and post-Intervention) and “Groups” (WB-EMS and Control) for primary outcomes, and “Time” (4 levels: pre- WB-EMS, post- WB-EMS, three, and six-month follow-up) and “Groups” (WB-EMS and Control) for secondary outcomes”.

4- Line 369: State the criteria that will be used to select the underlying covariance structure in the mixed linear model.

The percentage of missing data, effect size, and other non-normal distributions will be considered criteria for the covariance structures in the mixed linear model. The underlying covariance structure in the linear mixed model will be selected by minimizing the restricted maximum likelihood for the estimated models1,2.

References:

1. Wolfinger R. Covariance structure selection in general mixed models. Communications in Statistics - Simulation and Computation. 1993;22:4,1079-1106

2. Edmondson RN. Multi-level block designs for comparative experiments. Journal of Agricultural, Biological, and Environmental Statistics. 2020;25:4, 500–522

Reviewer #2: This manuscript describes the protocol of an 8-week RCT on the effect of whole-body electrical stimulation versus a resistance exercise program in older adults. The study includes a comprehensive set of measures of muscle mass, strength, and function, as well as secondary outcomes related to social participation and falls efficacy. The study protocol seems well thought-through, but I do have the following comments:

Major comments:

1) My main comment is about the aim of the study, based on the hypotheses (line 117): is it really expected that the ‘active control group’ will show greater reductions in strength and FFM than the WB-EMS group? I understand that the longer-term compliance to a RE program is not large, but I find it hard to believe that WB-EMS will have a greater effect on muscle function and strength than supervised resistance exercise. Would it not be better to compare the WB-EMS group with a true control group, doing habitual physical activity only?

Answer: Our hypothesis assumes that the WB-EMS will have a greater effect on strength than the supervised resistance exercise due to the following two reasons: First, the WB-EMS protocol is combined with low-load active exercise and the WB-EMS produces contractions triggered by electrical stimuli that leads to recruitment in random order compared with voluntary contraction. It is important to note that non-selective recruitment can provide clinical advantages, in which all fibers, regardless of type, have the potential to be activated at relatively low intensities1. Recalling that Type II fibers are those with the greatest capacity to generate strength and volume gains (i.e., muscle mass)2,3. Another important difference between volitional and non-volitional contractions (WB-EMS) is the synchronous activation of all motor units during contractions triggered by an electrical stimulus, which generates actin-myosin interaction at the same time in all recruited fibers. These factors can support the potential strength and mass gains compared with voluntary contraction during resistance exercise. Second, studies performed with the elderly and using the WB-EMS already included an inactive control group4,5. These points demonstrate the importance of knowing the potential superiority of WB-EMS in relation to conventional resistance exercise in the elderly. 

1. Gregory CM, Bickel CS. Recruitment patterns in human skeletal muscle during electrical stimulation. Phys Ther. 2005;85:358-364. 

2. Sjoholm H, Sahlin K, Edstrom L, Hultman E. Quantitative estimation of anaerobic and oxidative energy metabolism and contraction characteristics in intact human skeletal muscle in response to electrical stimulation. Clin Physiol, 1983; 3:227-239.

3. Vanderthommen M, Duteil S, Wary C, Raynaud JS, Leroy-Willig A, Crielaard JM, and Carlier PG. Comparison of isometric muscle training and electrical stimulation supplementing isometric muscle training in the recovery after major knee ligament surgery. A preliminary report. Am J Sports Med. 1979; 7: 169-71.

4. Mahoney ET., Bickel CS., Elder C., Black C., Slade JM., Apple D. Jr., et al.Changes in skeletal muscle size and glucose tolerance with electricallystimulated resistance training in individuals with chronic SCI. Arch Phys Med Rehabil 2005; 86: 1502-1504.

5. Gondin J., Guestte M., Jubeau M., Ballay Y., Martins A. Central and peripheral contribuition to fatiga after electrostimulation training. Med Sci Sports Exerc 2006; 38: 1147-1156.

2) What gap in the literature will this study fill? What will this study demonstrate that the studies in line 111 haven’t?

Answer: We added more information in this sentence:

“Studies involving WB-EMS in the elderly are scarce and assessed only structural, functional, or laboratory outcomes, such as body composition, strength, and lipid profile1,2,3. “No studies using WB-EMS involving older adults assessed the functional capacity outcomes related to the patient's fall efficacy and social participation or conducted a follow-up to evaluate the maintenance of the effect longitudinally” 

1. Kemmler W, Schlifka RJ, Mayhew L, Von Stengel S. Effects of whole-body electromyostimulation on resting metabolic rate, body composition, and maximum strength in postmenopausal women: The Training and ElectroStimulation Trial. J Strength Cond Res. 2010; 24(7):1880-7

2. Kemmler W, Bebenek M, Engelke K, Von Stengel S. Impact of whole-body electromyostimulation on body composition in elderly women at risk for sarcopenia: The Training and ElectroStimulation Trial (TEST-III). Age (Dordr). 2014; 36(1):395-406.

3. Kemmler W, Von Stengel S. Whole-body electromyostimulation as a means to impact muscle mass and abdominal body fat in lean, sedentary, older female adults: Subanalysis of the TEST-III trial. Clin Interv Aging 2013; 8:1353-64

3) The sample size calculation is currently unclear: is the study powered to pick up a change in variables in the WB-EMS group, or a difference between both intervention groups?

Answer: The reviewer is entirely right. Thank you for calling our attention to this issue. The sample size was calculated for all outcomes based on studies showing differences between groups in favor of EMS. However, there was a lapse for intra-group differences regarding the strength outcome. So, we recalculated the sample size for maximal voluntary contraction with a type-1 error of 0.05 and a power of 0.8 to detect a difference between groups. With an assumed effect size of 0.96 (Di Filippo 2017), an optimal sample size of 36 was obtained (18 participants per group). Please see below the sample size calculation using the G*Power software.

Reference: Di Filippo ES, Mancinelli R, Marrone M, Doria C, Verratti V, Toniolo L, Dantas JL, Fulle S, Pietrangelo T. Neuromuscular electrical stimulation improves skeletal muscle regeneration through satellite cell fusion with myofibers in healthy elderly subjects. J Appl Physiol (1985). 2017 Sep 1;123(3):501-512. doi: 10.1152/japplphysiol.00855.2016. Epub 2017 Jun 1. PMID: 28572500.

Minor comments:

1) With n=29 participants per group required for functional lower extremity strength, and a dropout of 5 participants per group, should the sample size not be n=34 per group?

Answer: We agree with you, and we already adjusted the number of participants. We considered the largest sample size calculated (i.e., 29 per group and added 15% for potential losses), resulting in 68 participants (34 per group).

2) Line 79: should this be ‘annual loss’?

Answer: Yes, you are right. We added the word “loss” in the sentence.

3) How will the four free options be decided? (line 218).

Answer: Although the WB-EMS device allows the increment of four free options, we chose not to use them, maintaining the stimulation of the muscle groups determined in the protocol. So, we deleted the information regarding the free four options presented in the Method section.

4) Line 307: time of day?

Answer: Yes, we added “of day” in the sentence.

5) Line 367: pre-intervention instead of pre-WB-EMS.

Answer: Thank you. We changed for “pre-intervention”.

6) Line 398: why poor adherence to long-term programs relevant for such a short study?

Answer: We agree that this assertion does not make sense. So, we removed the word “long-term”.

Reviewer #3: Thank you for the opportunity to review this manuscript. In general, it is well written and well designed.

I have just a few comments regarding the methods.

1- I believe that the word “sex” is more appropriated than “gender”.

Answer: We have rewritten with the word “sex”, as suggested.

2- The randomization method should be better described. The authors describe the allocation concealing method with opaque envelopes, which is appropriated. However, how will the random allocation be performed? Will the authors use some software or website for this? Or will the authors draw envelopes? Please clarify.

Answer: We have added information about the randomization method using the “Random Allocation Software (RAS)”, and we also provided detail about the allocation sequence, which will be conducted in identical sequentially numbered, opaque, and sealed envelopes. The randomization will be concealed until the end of the evaluations.

3- “Pulse width” should be replaced by “pulse duration” as it is measured in time unit.

Answer: According to your suggestion, we have replaced the “pulse width” for “pulse duration”.

4- The current amplitude/intensity adjust should be better described. Please give more details. Will the amplitude be increased during the same treatment session?

Answer: According to your observation, we have added the following information: “The current intensity will be selected individually and modified gradually according to self-perceived discomfort threshold during the EMS session.”

5- Why the ON time will be longer than the OFF time 6/4? In traditional NMES programs for increase of muscle strength the OFF time is longer than the ON time to avoid fatigue. Please clarify.

Answer: We appreciate your observation. Our ON/OFF protocol was based on studies using the WB-EMS1,2. Nevertheless, we agree with your observation, and adjusted for a 1:2 ratio (5 sec On and 10 sec OFF).

References:

1. Kemmler W, Grimm A, Bebenek M, Kohl M, von Stengel S. Effects of Combined Whole-Body Electromyostimulation and Protein Supplementation on Local and Overall Muscle/Fat Distribution in Older Men with Sarcopenic Obesity: The Randomized Controlled Franconia Sarcopenic Obesity (FranSO) Study. Calcif Tissue Int. 2018 Sep;103(3):266-277

2. Kemmler W, Weissenfels A, Teschler M, Willert S, Bebenek M, Shojaa M, Kohl M, Freiberger E, Sieber C, von Stengel S. Whole-body electromyostimulation and protein supplementation favorably affect sarcopenic obesity in community-dwelling older men at risk: the randomized controlled FranSO study. Clin Interv Aging. 2017 Sep 21;12:1503-1513.

6- Why did authors decided to perform 2 sessions per week rather than 3? Are 2 sessions per week an appropriated dose for exercise/electrical stimulation?

Answer: Our group is conducting a systematic review with the same purpose (please see the following registration at PROSPERO database (CRD42019134100) https://www.crd.york.ac.uk/prospero/display_record.php?RecordID=134100”

In general, all trials1,2.3 that used this wearable technology in the elderly provided treatment once or twice a week. Therefore, we rely on these references to choose the number of sessions offered in the protocol. Also, considering that the whole body non-volitional stimulus (WB-EMS) is different from the volitional stimulus (resistance exercise), we believe that it is not mandatory to match the number of sessions for an equivalence stimulus between the two interventions.

References:

1. Kemmler W, Grimm A, Bebenek M, Kohl M, von Stengel S. Effects of Combined Whole-Body Electromyostimulation and Protein Supplementation on Local and Overall Muscle/Fat Distribution in Older Men with Sarcopenic Obesity: The Randomized Controlled Franconia Sarcopenic Obesity (FranSO) Study. Calcif Tissue Int. 2018 Sep;103(3):266-277

2. Kemmler W, Teschler M, Weissenfels A, Bebenek M, von Stengel S, Kohl M, Freiberger E, Goisser S, Jakob F, Sieber C, Engelke K. Whole-body electromyostimulation to fight sarcopenic obesity in community-dwelling older women at risk. Results of the randomized controlled FORMOsA-sarcopenic obesity study. 2016 Nov;27(11):3261-3270. 

3. Wittmann K, Sieber C, von Stengel S, Kohl M, Freiberger E, Jakob F, Lell M, Engelke K, Kemmler W. Impact of whole body electromyostimulation on cardiometabolic risk factors in older women with sarcopenic obesity: the randomized controlled FORMOsA-sarcopenic obesity study. Clin Interv Aging. 2016 Nov 18;11:1697-1706.

7- In the Primary and Secondary Outcomes section the timepoint (s) for each outcome should be described.

Answer: According to your recommendation, we added information about the time points for the primary and secondary outcomes (see Methods – section Primary Outcome and Secondary Outcome).

8- I suggest some assessment regarding patient’s satisfaction. In introduction authors describe the lack of adherence in exercise programs and suggest the whole-body electromyostimulation as a new training strategy to improve older people’s adherence to the therapeutic programs. Therefore, I believe it is very important to assess the patients’ impressions on this new method compared to the traditional (control group).

Answer: We agree with you and we have added the instrument “Patients’ satisfaction with and awareness of electrical stimulation therapy” to assess the patient’s satisfaction.

This questionnaire includes 2 sections. Section 1, consisting of 10 items, addressed demographic details such as age, gender, education level, application body areas, number of treatment sessions, electrical stimulation therapy (EST) modalities, perceived healing effect, devices, and probes of EST. Data will be collected via closed-ended categorical and yes/no questions. Section 2, consisting of 3 items, investigated the subjects’ having information on EST, their knowledge of EST, and their satisfaction. The questionnaire will be administered face to face to the volunteers.

Reference:

Kocamaz D, Yakut H, Özberk S. Patients’ satisfaction with and awareness of electrical stimulation therapy. Physiotherapy Quarterly. 2020, 28(1), 11–15.

---

## [Decision Letter · Decision Letter 1]

8 Jan 2021

Effects of whole-body electromyostimulation on function, muscle mass, strength, social participation, and falls-efficacy in older people: A randomized trial protocol

PONE-D-20-23831R1

Dear Dr. MALAGUTI,

We’re pleased to inform you that your manuscript has been judged scientifically suitable for publication and will be formally accepted for publication once it meets all outstanding technical requirements.

Kind regards,

Cameron J. Mitchell, PhD

Academic Editor

PLOS ONE

Additional Editor Comments (optional):

Reviewers' comments:

Reviewer's Responses to Questions

**Comments to the Author**

1. Does the manuscript provide a valid rationale for the proposed study, with clearly identified and justified research questions?

Reviewer #1: Yes

Reviewer #2: Yes

2. Is the protocol technically sound and planned in a manner that will lead to a meaningful outcome and allow testing the stated hypotheses?

Reviewer #1: Yes

Reviewer #2: Yes

3. Is the methodology feasible and described in sufficient detail to allow the work to be replicable?

Reviewer #1: Yes

Reviewer #2: Yes

4. Have the authors described where all data underlying the findings will be made available when the study is complete?

Reviewer #1: Yes

Reviewer #2: Yes

5. Is the manuscript presented in an intelligible fashion and written in standard English?

Reviewer #1: Yes

Reviewer #2: Yes

6. Review Comments to the Author

You may also provide optional suggestions and comments to authors that they might find helpful in planning their study.

Reviewer #1: All comments have been adequately addressed.

Reviewer #2: The authors have satisfarily addressed all my comments, and I wish them the best of luck with the study.

7. PLOS authors have the option to publish the peer review history of their article (what does this mean?). If published, this will include your full peer review and any attached files.

Reviewer #1: No

Reviewer #2: No

---

## [Editor Report · Acceptance letter]

15 Jan 2021

PONE-D-20-23831R1 

Effects of whole-body electromyostimulation on function, muscle mass, strength, social participation, and falls-efficacy in older people: A randomized trial protocol 

Dear Dr. Malaguti:

I'm pleased to inform you that your manuscript has been deemed suitable for publication in PLOS ONE. Congratulations! Your manuscript is now with our production department. 

Kind regards, 

on behalf of

Dr. Cameron J. Mitchell 

Academic Editor

PLOS ONE